**Data Availability Statement:** The data collected for this study are available at the UK Data Service (doi:

# Common and distinct predictors of non-symbolic and symbolic ordinal number processing across the early primary school years

Sabrina Finke[1]*, Chiara Banfi[1,2], H. Harald Freudenthaler[1], Anna F. Steiner[1,3], Stephan E. Vogel[1], Silke M. Göbel[4,5], Karin Landerl[1,6,7]

1 Institute of Psychology, University of Graz, Graz, Austria, 2 Institute for Medical Informatics, Statistics and Documentation, Medical University of Graz, Graz, Austria, 3 FH JOANNEUM, University of Applied Sciences, Graz, Austria, 4 Department of Psychology, University of York, York, United Kingdom, 5 Department of Special Needs Education, University of Oslo, Oslo, Norway, 6 BioTechMed-Graz, Graz, Austria, 7 Department of Cognitive Science, Macquarie University, Sydney, NSW, Australia

* sabrina.finke@uni-graz.at

## Abstract

What are the cognitive mechanisms supporting non-symbolic and symbolic order processing? Preliminary evidence suggests that non-symbolic and symbolic order processing are partly distinct constructs. The precise mechanisms supporting these skills, however, are still unclear. Moreover, predictive patterns may undergo dynamic developmental changes during the first years of formal schooling. This study investigates the contribution of theoretically relevant constructs (non-symbolic and symbolic magnitude comparison, counting and storage and manipulation components of verbal and visuo-spatial working memory) to performance and developmental change in non-symbolic and symbolic numerical order processing. We followed 157 children longitudinally from Grade 1 to 3. In the order judgement tasks, children decided whether or not triplets of dots or digits were arranged in numerically ascending order. Non-symbolic magnitude comparison and visuo-spatial manipulation were significant predictors of initial performance in both non-symbolic and symbolic ordering. In line with our expectations, counting skills contributed additional variance to the prediction of symbolic, but not of non-symbolic ordering. Developmental change in ordering performance from Grade 1 to 2 was predicted by symbolic comparison skills and visuo-spatial manipulation. None of the predictors explained variance in developmental change from Grade 2 to 3. Taken together, the present results provide robust evidence for a general involvement of pair-wise magnitude comparison and visuo-spatial manipulation in numerical ordering, irrespective of the number format. Importantly, counting-based mechanisms appear to be a unique predictor of symbolic ordering. We thus conclude that there is only a partial overlap of the cognitive mechanisms underlying non-symbolic and symbolic order processing.

10.5255/UKDA-SN-854335). Data of test-retest reliability for the ordinal processing tasks are publicly available on the Open Science Framework at https://osf.io/gm498/ (doi: 10.17605/OSF.IO/GM498).

**Funding:** This work was supported by the Austrian Science Fund (www.fwf.at), grant number I 2778-G16 awarded to KL, and the Economic and Social Research Council (www.esrc.ukri.org), grant number ES/N014677/1 awarded to SMG. The funders had no role in study design, data collection and analysis, decision to publish, or preparation of the manuscript.

**Competing interests:** The authors have declared that no competing interests exist.

# Introduction

There is a growing interest in how the mind represents numbers. Previous studies on number processing primarily focused on cardinality, which refers to the numerical property that qualifies the quantity of a set. Cardinality relates to the question "How many?" and it always involves absolute quantities ("five dots"). While cardinality has been extensively investigated at the cognitive and neuronal level (e.g., [1]), far less is known about another important numerical characteristic: How are numbers represented as part of ordered sequences? Ordinality refers to the relative position or rank of a number in a given sequence (e.g., in the sequence 1-2-3, 2 comes after 1 but before 3). A growing body of evidence has revealed the unique predictive contribution of ordinal processing skills to arithmetic performance in children and adults (e.g., [2–6]). Despite its importance for arithmetic, surprisingly little is known about the cognitive underpinnings of order processing. Importantly, the cognitive mechanisms may differ depending on the number format of these sequences, that is, whether the sequences are presented non-symbolically (e.g., sequences of dots; ●-●●-●●●) or symbolically (e.g., sequences of visual-Arabic digits; 1-2-3) [7,8]. It is also currently not known how the contributions of distinct cognitive mechanisms change during development. To this aim, the current study addresses the unique predictive contribution of non-symbolic and symbolic magnitude processing, and counting to the development of non-symbolic and symbolic order processing. We also investigated the relevance of more general cognitive mechanisms of verbal and visuo-spatial working memory.

## Non-symbolic vs. symbolic ordering

Ordinal processing tasks typically consist in judging whether a numerical sequence is in order (e.g., 1-2-3) or not in order (e.g., 1-3-2). Importantly, behavioral response time patterns differ depending on the number format of the ordered sequence, that is, whether the sequence consists of symbolic numbers (e.g., sequences of Arabic digits; 1-2-3) or of non-symbolic numbers (e.g., e.g., sequences of dots; ●-●●-●●●). For symbolic ordering, a reversed distance effect was reported [9], meaning that participants were faster to judge that a pair ascending by one (e.g., 3–4) is in order than a pair ascending by more than one (e.g., 3–5). This was a seminal finding, considering the fact that the opposite (canonical distance effect) is reliably observed for pairwise magnitude comparison tasks [10]: Participants are faster when judging which of two numbers is larger in a pair with large numerical distance (e.g., 1–9) than in a pair with a small numerical distance (e.g., 1–2). Reversed distance effects have since been replicated for symbolic ordering [8,11,12]. Recently, it has been suggested that this facilitation in response time for consecutive items points towards an efficient retrieval of learned ordered sequences from long-term memory [3,7,13–15].

Critically, for non-symbolic order judgements only canonical distance effects could be found, whereas reversed distance effects were entirely absent [5,8]. This suggests that non-symbolic order processing may be strongly reliant on multi-stage magnitude comparisons. For example, for the triplet ●-●●-●●●, an individual may first compare the pair of dots on the left and decide that two is larger than one. The pair of dots on the right is compared in a second step, leading to the decision that three is larger than two. Finally, the full triplet is judged as being in ascending order. The fact that only canonical distance effects can be observed in non-symbolic ordering has led researchers to propose that non-symbolic order processing may be closely related to cardinal processing of non-symbolic magnitudes. In contrast, the existence of reverse distance effects for directly ascending symbolic items may be caused by direct retrieval from the verbal count list (e.g., "1–2" is part of the counting sequence "one-two"), pointing towards the engagement of retrieval-based mechanisms in symbolic order processing [4,7,13].

Further evidence for the involvement of different mechanisms in non-symbolic and symbolic order processing comes from functional magnetic resonance imaging (fMRI) studies with adult participants, showing differential brain activity related to symbolic and non-symbolic number formats [8,16]. One fMRI study [8] examined the neural signatures of cardinal and ordinal processing of symbolic numbers (visual-Arabic digits) and non-symbolic quantities (dots), as well as a non-numerical control condition (luminance). For all numerical tasks, stimuli consisted of numerosities ranging from 1 to 9. The non-symbolic and symbolic cardinal processing tasks consisted of magnitude comparison tasks with dots and digits, respectively. In both non-symbolic and symbolic ordinal processing tasks, participants were required to judge whether triplets of stimuli were ordered from left to right (increasing or decreasing) or not in order. Numerical processing was determined by subtracting brain activation associated with the control condition from each numerical task. There was a strong overlap of the neural networks involved in cardinal and ordinal processing of non-symbolic quantities: Both cardinal and ordinal judgements were associated with activations of a right-lateralized fronto-parietal network, including the dorsolateral prefrontal cortex, the intraparietal sulcus and the anterior cingulate cortex. In contrast, the authors could not find any overlap in the brain activation for ordinal and cardinal processing of symbolic quantities. These results suggest a tight link between cardinal and ordinal processing of non-symbolic numbers, while such a link is less obvious for symbolic numbers. Of note, symbolic ordinal processing was selectively associated with activation of premotor regions in the left hemisphere.

A recent study investigated the similarity between patterns of neural activation evoked by non-symbolic and symbolic quantities using a delayed match-to-sample task in adults [16]. Participants were required to indicate whether pairs of dot arrays or digits ranging between 1 and 9 that were presented with a jittered delay were identical. The authors considered brain activity during the presentation of the first stimulus, as well as during the delay before the onset of the second stimulus. Representational similarity analysis was conducted to compare brain activation patterns for non-symbolic and symbolic quantities. Results showed that brain activation for non-symbolic quantities depended on numerical ratio in prefrontal, parietal and occipital areas. In contrast, brain activation for symbolic quantities was unrelated to numerical ratio, but showed an association with lexical frequency. These differential patterns suggest that processing of non-symbolic and symbolic numbers is supported by distinct and largely independent cognitive mechanisms.

In summary, preliminary evidence indicates that the cognitive mechanisms supporting symbolic ordering (e.g., sequences of Arabic digits; 1-2-3) are partly different from the mechanisms involved in non-symbolic ordering (e.g., sequences of dots; ●-●●-●●●) [14], at least in adults.

## Developmental predictors of numerical order processing

Notably, the neurocognitive mechanisms supporting the development of non-symbolic ordinal processing in children have been largely neglected. In contrast to non-symbolic ordering, some research has been conducted on the developmental foundations of symbolic ordering. In the next section we discuss preliminary evidence, which mainly highlights the interplay between cardinal and ordinal magnitude comparison, the contribution of retrieval-based mechanisms such as counting, and the involvement of different working memory components.

**Cardinal magnitude processing.**    Developmental studies have pointed towards a complex and interactive link between numerical order and numerical magnitude [1]. Cross-sectional evidence indicates that the interplay between the ability to process the order and the

magnitude of digits changes as a function of age, which is also relevant for the prediction of arithmetic performance: In Grade 1, digit comparison mediated the relationship between symbolic order processing and arithmetic, whereas in Grade 2, the opposite pattern was observed, with symbolic order processing mediating the predictive role of digit comparison for arithmetic [15]. This finding indicates that the two dimensions might interact when children form precise mental representations of symbolic number [17,18]. However, the direction of this relation is still unclear: There is evidence that numerical order knowledge plays a role for numerical magnitude judgements [19], and there is also evidence that numerical magnitude knowledge drives the development of order processing [15]. Arguably, the direction of the causal relation may change during the first years of formal schooling, because children are gaining increasing proficiency with the Arabic number system. Based on preliminary cross-sectional evidence [15], the development of order processing appears to be highly dynamic at the beginning of primary school, and cardinal processing may constitute a driving factor for its improvement. Of note, it is possible that the extent of cardinality-based mechanisms involved in order processing may critically depend on whether the stimuli are non-symbolic or symbolic representations of quantities [14].

Consequently, the current study aims to investigate the contribution of cardinal magnitude processing to non-symbolic and symbolic order processing at the beginning of primary school. Adopting a longitudinal design allows us to address the relation at the beginning of formal schooling in Grade 1, and even more importantly, we will be able to investigate the contribution of magnitude processing skills to developmental change in order processing over the first three years of primary school.

**Counting.** A candidate mechanism with potentially stronger contribution to symbolic than to non-symbolic ordering is the ability to retrieve overlearned ordinal information from the verbal count-list [14,20,21]. From a developmental perspective, establishing a reliable link between exact quantities and symbolic numbers in early childhood has been proposed to be based on counting skills [22]. Children are often able to recite the list of count words before they develop a full understanding of one-to-one correspondence and cardinality inherent in counting procedures [18,23]. Longitudinal evidence revealed that preschoolers' ability to deal with symbolic numbers greatly improves once they have mastered the counting principles, whereas this is not the case for non-symbolic numbers [24]. Direct retrieval of the over-learnt count-list offers a plausible explanation for the reversed distance effects observed in symbolic ordering: Participants may be faster to decide that 1–2 is ordered compared to 1–3, because the former is part of the verbal counting chain in long-term memory, whereas the latter is not. This is supported by recent work [4,11] showing that adult participants were fastest when judging symbolic triplets at the lower end of the count list such as 1-2-3. A cross-sectional study using a dot enumeration task did not find the expected association between counting skills and symbolic order processing in a sample of children in Grades 1 to 6 [20]. However, enumerating dot numbers between 1 and 9 is a complex task in cognitive terms because it involves not only knowledge of the count list, but also efficient application of the one-to-one principle of counting. For smaller dot sets up to four dots, the participants of that study may have applied subitizing skills, i.e., a visually based parallel process that does not involve the sequential verbal count list [25]. To clarify the contribution of counting to order processing, the present study employed a more basic measure of rote counting, requiring participants to orally recite the count list.

In summary, there is reason to assume that counting uniquely contributes to symbolic order processing, over and above the cognitive mechanisms involved in non-symbolic order processing. Since counting is already well-developed in children at the start of their formal education [18], and more sophisticated calculation strategies come into use through formal

schooling [26], the contribution of counting to order processing may diminish from Grade 1 to Grade 3. At the same time, children's familiarity with exact symbolic quantities develops rapidly during the first years of schooling, as this constitutes the main focus of mathematics instruction [27]. A plausible assumption is that the efficiency of symbolic magnitude processing may be one of the determinants of developmental change in order processing during this period. Longitudinal evidence is crucial to determine whether children's symbolic magnitude processing skills at the beginning of primary school have an impact on the development of their ordering skills in the subsequent years.

**Working memory.**    Working memory allows individuals to store and manipulate a limited amount of information for short periods of time, and it plays a major role for various forms of numerical processing [28–30]. Importantly, working memory should not be viewed as a monolithic construct, but instead consists of specific components for processing verbal and visuo-spatial information [31,32]. Apart from the differentiation between specific sensory domains, an important issue pertains to the distinction between storage and manipulation: Is information only retained in memory (storage) or are additional processing steps involved (manipulation)? On a theoretical level, it appears highly reasonable to assume a contribution of memory components to order processing by storing and manipulating task-relevant information throughout multiple processing steps [33–35].

Verbal working memory may play a prominent role for symbolic ordering: When making order judgements, individuals may retrieve the verbal labels (number words) corresponding to the visually presented constituents of a sequence (digits) in order to compare these to the count-list stored in long-term memory. For instance, when processing the sequence 1-2-3, the corresponding verbal number words one-two-three are retrieved. Verbal storage is relevant for order processing because it enables maintaining the verbal representations of a given sequence, which are in turn compared to the representations stored in verbal long-term memory. This notion is supported by the phenomenon that more familiar triplets (e.g., 1-2-3) are more easily judged than unfamiliar ones (e.g., 2-4-6): Adults were shown to process numerical order items with a distance of one, especially ascending items, in a highly automatic fashion [4]. This finding could since be replicated in a study that employed a series of experiments manipulating various numerical and non-numerical characteristics of ordered sequences [11]. Similar to [4], participants were especially fast when judging small triplets with a small numerical distance between the constituents. The authors argued that these small and consecutive items occur more frequently in every-day language, resulting in a more efficient storage and retrieval.

The manipulation component of verbal working memory may also play a relevant role for order processing, especially for sequences that are not in order: Besides short-term maintenance, the verbal representation of these sequences may need to be manipulated in order to compare them to the verbal count-list stored in long-term memory. Indeed, previous studies with children [36] and adults [37] reported significant associations between symbolic order processing and performance in complex verbal working tasks involving both manipulation and storage across multiple trials. To disentangle between the contributions of storage and manipulation components of verbal working memory, we argue that it is important to test whether the manipulation component can uniquely contribute to the prediction of order processing over and above verbal storage.

Although there is convincing evidence for a significant relation between visuo-spatial working memory and arithmetic [38–40], a potential involvement of visuo-spatial working memory components in ordering tasks has been largely neglected. Only recently, evidence [4,41] has emerged pointing towards an involvement of visuo-spatial processing in order processing. One study reported an ordinal Stroop paradigm in which participants had to judge the

physical size of numerically ascending, descending and not-in-order triplets [4]. Participants were faster to indicate that the physical size of number triplets that were in-order when the numerical and physical values were congruent, but interestingly, this facilitation effect was only found for trials with a left-to-right, ascending physical orientation. Thus, fast visual recognition may play a role in order processing, and the efficient retrieval of ordered sequences may depend on the mental representation of their spatial properties. In a similar vein, others proposed that visuo-spatial memory may be an important prerequisite for learning the spatial-ordinal relations of digits [41]. Testing this account in a sample of preschool children, order processing was assessed by means of a task requiring participants to correctly place digits to the left or right side of a reference number on a number line. The authors reported a significant association between ordering and children's performance on a visuo-spatial storage task involving recall of form and location of abstract objects. Thus, it appears reasonable to assume an involvement of the storage component of visuo-spatial working memory in ordering, at least when symbolic numbers are involved.

Moreover, if it is true that (at least non-symbolic) ordering tasks are solved by pair-wise magnitude comparisons [8,14], visuo-spatial manipulation skills might play a role when splitting ordered triplets into two sequential magnitude comparison tasks. We therefore assume that children with better visuo-spatial manipulation skills are at an advantage when solving order processing tasks.

Findings on the role of different working memory components for numerical processing in general and order processing in particular are still inconclusive, as many empirical studies only assessed one of these components or could not clearly dissociate between different components [38,42]. The latter is particularly the case for serial order working memory tasks, which were previously shown to be related to symbolic order processing in children [43]. As pinpointed by previously [38], serial order processing tasks involve both verbal and visuo-spatial working memory processes: Participants are required to retain a series of auditorily presented words (verbal memory) before arranging visual cards corresponding to these words spatially from left to right in the same order that they were presented (spatial and verbal memory). In summary, it remains necessary to disentangle the relation between order processing and "pure" measures of verbal and visuo-spatial storage and manipulation.

In this study, we assessed working memory ability by means of the classic paradigms of digit span and block tapping, which allows us to differentiate between storage and manipulation components of verbal and visuo-spatial working memory, respectively.

## The present study

The central aim of the present study was to investigate the common and distinct numerical mechanisms supporting non-symbolic and symbolic order processing in primary school children. In line with extant evidence as well as recent theoretical accounts suggesting distinct neurodevelopmental trajectories of symbolic and non-symbolic number knowledge [23,44,45], we expected a particularly strong predictive contribution of non-symbolic magnitude processing skills to non-symbolic order processing. Similarly, it is reasonable to assume a format-dependent association between symbolic magnitude processing and symbolic order processing. Employing a hierarchical approach, we aimed to determine whether counting would account for unique variance in symbolic order processing over and above magnitude comparison skills and working memory. We also investigated the contribution of specific components of working memory to order processing and its development. In particular, we aimed to disentangle the influence of storage and manipulation components of verbal and visuo-spatial working memory on non-symbolic and symbolic ordering performance.

In a longitudinal study, we repeatedly assessed children´s non-symbolic and symbolic order processing skills in Grades 1, 2, and 3. This design enabled us to determine the stability of the association between non-symbolic and symbolic order processing over a period in which children gain increasing proficiency with the symbolic number system. The cognitive determinants of ordinal processing were analyzed concurrently in Grade 1, as well as longitudinally by predicting developmental change in order processing.

We employed a timed paper-and-pencil adaptation of a computerized ordinal processing task [7], which is particularly useful to measure order processing with groups of children. This task format has successfully been used for non-symbolic and symbolic magnitude comparison paradigms in a number of studies [46,47], including the current one. Due to the speeded nature of the ordering and magnitude processing tasks, we decided to control for processing speed in all analyses.

## Materials and method

### Participants

The present sample consisted of 157 children (75 females) with a mean age of 7.15 years ($SD$ = 0.29) at the first assessment time point. Initially, 177 native German-speaking children from five primary schools (12 classrooms) in an urban school district in Austria took part in our three-year longitudinal study starting at the end of Grade 1. We had to exclude 11 children who did not take part in all three assessment timepoints from Grade 1 to 3. We further excluded nine children who showed a clearly biased answer pattern in either of the ordinal processing tasks (i.e., ticking or crossing out more than 10 items in a row).

This study was approved by the ethics committee of the University of Graz (case identification code: 39/23/63 ex 2016/17), and written informed consent was granted by the parents or legal guardians.

### Power

We determined the appropriate sample size for regression models with seven predictors based on the convention of a minimum of $N = 104 + m$ (number of predictors) for regression models [48]. Due to the three-year longitudinal design of our study, we were concerned about the problem of attrition and therefore decided to invite a larger number of participants to take part in the study. We also conducted a sensitivity analysis in G*power [49], setting power to .80 and the probability of alpha-error to .05. This indicated that we would be able to detect an incremental effect of $\triangle R^2$ = .05 with the present sample size of N = 157.

### Tasks

**Non-symbolic order processing.**   Children were asked to decide whether three displays of black dots in a row were in ascending numerical order (e.g., 2–4–6 dots) or not in order (e.g., 2–6–4 dots), see Fig 1 for example items. Descending items were not included, because piloting revealed that such a task version was too difficult for children in Grade 1. Children were presented with an A4 booklet with eight pages. Each page contained two columns with five items each, resulting in a total of 80 items. Stimuli consisted of three numerosities between 1 and 9. Altogether, 41 ascending triplets were included (between 3 and 6 per page). The remaining triplets were not in order. To ensure that magnitude was more salient than the physical features of the stimuli throughout the task, the overall surface area of the dots was either correlated or anti-correlated with the number of dots (i.e., surface area either increased or decreased with the number of dots). Numerical distance between the three sets of numerosities

a)

b)

c)

d)

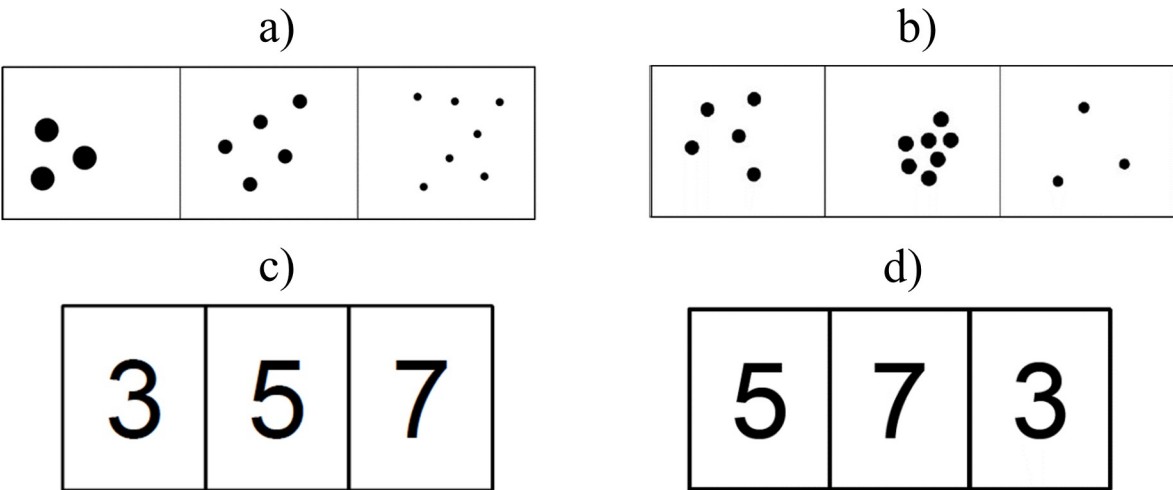

**Fig 1.** Example items of the non-symbolic and symbolic ordering tasks: non-symbolic a) ascending and b) not-in-order items, and symbolic c) ascending and d) not-in-order items. Children were asked to indicate whether magnitudes were presented in an ascending order or not.

in the ordered condition was either one (e.g., 1-2-3 dots), two (e.g., 1-3-5 dots) or three (e.g., 1-4-7 dots). Children were instructed to tick items with triplets presented in an ascending order and to cross out items in which the number sequence was not in order. They were specifically instructed to work as fast as possible, without making mistakes. Explicit counting was discouraged. Before starting the actual tasks, the experimenter presented and discussed four practice items and made sure that children understood the instruction. The number of correct responses given within a time limit of 90s was scored.

Test-retest reliability from a separate sample of 25 2nd graders appeared moderate at first, with $r(23) = .57$, $p < .001$. Closer inspection showed that two children of this reliability sample showed clearly lower performance on the second execution of the task, while all other children improved task performance in terms of a familiarization effect, so we suspected that these two outliers did not perform the task properly. After outlier exclusion, reliability was, $r(21) = .82$, $p < .001$. (Data of test-retest reliability are available on the Open Science Framework at https://osf.io/gm498/.)

**Symbolic order processing.** The task format was analogous to the non-symbolic ordering condition, but digits were presented instead of dots (Fig 1). Altogether, 35 ascending triplets were presented (between 2 and 6 per page) among the 80 items on 8 pages. Before starting the actual tasks, the experimenter completed six practice items together with the children and ensured that they understood the instruction. The number of correct responses given within a time limit of 90s was scored. Test-retest reliability from the same sample of 25 2nd graders on which the reliability for the non-symbolic ordering condition was based showed a high correlation of $r(23) = .74$, $p < .001$.

**Processing speed.** We employed a symbol cancellation task as a measure of processing speed. Children were given an A5 booklet with one practice and 12 test pages, each presenting six rows with a star and a circle. The star was randomly located on the left or the right side. Children were asked to tick the stars, but not the circles as fast as possible. Processing speed was measured as the number of items marked correctly within 30 seconds.

*Working memory.* Verbal working memory was assessed with the Digit Span subtests from the Working Memory Test Battery for Children (WMTB-C, [50]). As a measure of verbal *storage*, we employed the Digit Span Forward task (test-retest reliability is .81 [51]). The

experimenter read out a series of digits of increasing length and children were required to repeat this sequence in the same order. Digit Span Backward additionally requires the *manipulation* component of verbal WM (test-retest reliability is .53 [51]). The experimenter read out a series of digits of increasing length and children were asked to repeat the series of digits in reverse order. Four practice trials preceded the administration of the test trials.

**Visuo-spatial working memory.** The *storage* component of visuo-spatial working memory was assessed with the Block Recall Forward subtest of the Working Memory Test Battery for Children (WMTB-C, [50]; test-retest reliability is .63, [51]). Children were presented with a set of nine identical blocks arranged on a board. The experimenter tapped a block sequence and children were asked to tap the same blocks in the same order. To measure the *manipulation* component of visuo-spatial working memory, we added a backward block tapping condition. The experimenter tapped a block sequence and children were asked to tap the same blocks in the reversed order. Two practice items preceded the administration of the test trials.

For all working memory tasks, six items were included per span length. If a child correctly recalled four out of six items of a span, he or she proceeded directly to the next span and was given credit for any omitted trial (move on rule). The task was discontinued as soon as there were three incorrect items within a certain span (discontinuation rule). Within the final span, correctly solved items were scored until the discontinuation rule came into effect. The raw score corresponded to the total amount of correct sequences.

**Non-symbolic magnitude comparison.** Children completed non-symbolic magnitude comparison tasks as reported in a previous study [47]. Six pairs of items were presented on each page of an A5 booklet. Children were instructed to process as many pairs as possible in 30 seconds by ticking the numerically larger quantity in each pair. They were explicitly instructed not to count. Four different tasks were given, preceded by an additional practice task. All tasks involved pairs of arrays containing 5 to 40 black squares presented in a box. Two tasks included different numbers of equally sized squares, ranging between 5 and 13 per array. In one of these same-size tasks the numerical distance between the pairs of arrays was small (one or two). The other same-size task involved pairs of arrays with a large numerical distance (five, six, or seven). The other two non-symbolic task versions controlled for surface area of the squares to prevent judgements being made based on surface area or "blackness" of the arrays. These same-surface-area tasks included arrays of between 20 and 40 squares, differing by a certain ratio (3:4 and 5:6). The non-symbolic magnitude comparison score was calculated by averaging *z*-scores of the total correct scores across the four booklets. Cronbach's alpha reliability for the non-symbolic magnitude comparison task was .75.

**Symbolic magnitude comparison.** The task format was analogous to the non-symbolic magnitude comparison task but involved two tasks presenting digits instead of dots. Both tasks comprised single-digit numbers from 1 to 9. We included two task versions in order to control for numerical distance: While one task included number pairs with a small numerical distance (one or two) between both digits, the other task consisted of pairs with a large distance (five, six, or seven). The symbolic magnitude comparison score was calculated by averaging *z*-scores of the total correct scores of both booklets. Cronbach's alpha for the symbolic magnitude comparison task was .75.

**Counting speed.** Children recited the verbal count-list with a time limit of 60 seconds. They were asked to count as fast and accurately as possible starting with "1". Counting speed was assessed as the number of correctly recited numbers per second. If a child made more than three mistakes, counting speed was assessed as the number of correctly recited numbers before the third mistake, divided by 60.

## Procedure

Children performed the tasks as part of a larger battery in a longitudinal project. The dependent measures of ordinal processing were assessed at three time points: towards the end of Grade 1 and about one and two years later in Grades 2 and 3. Task order was fixed across all timepoints. Children completed the magnitude comparison tasks before receiving the non-symbolic and symbolic order processing tasks. Predictors were assessed once in Grade 1 with the exception of processing speed, which was assessed in Grade 2. Based on evidence showing high stability of processing speed performance in childhood [52,53] it was nevertheless considered as a control/predictor measure in this study. Ordinal processing, magnitude comparison, as well as processing speed were administered in a classroom setting. Verbal and visuo-spatial working memory and counting speed were assessed individually in a quiet room in the participating schools.

## Predicting ordinal processing and its development

We performed hierarchical linear regression analyses predicting non-symbolic and symbolic order processing in Grade 1. To control for any influence of processing speed, we entered processing speed in the first step of the model. In a second step, we entered performance in the digit span forward and block recall forward tasks as measures of the storage components of verbal and visuo-spatial working memory. Entering the backward task versions in the third step allowed us to examine whether the manipulation components of working memory contributed additional variance to the prediction of ordinal processing. In a fourth step, we introduced non-symbolic and symbolic magnitude comparison to the prediction. In a fifth step, we entered counting to test for a specific contribution of retrieval-based mechanisms to symbolic ordering. This allowed us to test whether counting uniquely predicted symbolic order processing over and above other cognitive skills (i.e., non-symbolic and symbolic magnitude comparison, working memory measures and processing speed).

We performed further linear regression analyses to predict developmental change in symbolic and non-symbolic order processing between Grades 1 and 2 and Grades 2 and 3. We quantified developmental change in order processing in terms of residualized change scores, providing a particularly useful measure of change when there is a high correlation or causal effect between earlier and later measures [54]. For instance, to calculate change in non-symbolic order processing from Grade 1 to Grade 2, we regressed non-symbolic ordering at Grade 2 on non-symbolic ordering at Grade 1. These residualized change scores capture variance that can be attributed to change in order processing between two timepoints. The residual is the unexplained variance in later ordinal processing skills after the effect of earlier ordinal processing skills has been partialled out. (For standard error and confidence intervals of the regression coefficients, see supporting information, S2 and S3 Tables).

## Results

The data collected for this study are available at https://reshare.ukdataservice.ac.uk/854398/.

## Development of order processing

Descriptive statistics of order processing at all three assessment timepoints are reported in Table 1. As expected, performance in non-symbolic and symbolic ordinality tasks increased across Grades 1 to 3, with moderate stability across grades (non-symbolic order processing Grade 1 –Grade 2 and Grade 2 –Grade 3: $r(155) = .49$, $p < .001$; symbolic order processing Grade 1 –Grade 2: $r(155) = .59$ and Grade 2 –Grade 3: $r(155) = .71$, both $p < .001$). Concurrent

**Table 1. Means (standard deviations in parentheses) of correctly solved items in the non-symbolic and symbolic order processing tasks in Grade 1, Grade 2, and Grade 3.**

|  | Grade 1 | Grade 2 | Grade 3 |
|---|---|---|---|
|  | M *(SD)* | M *(SD)* | M *(SD)* |
| **Non-symbolic ordering** | 11.82 (4.99) | 15.78 (5.45) | 20.22 (6.19) |
| **Symbolic ordering** | 19.16 (6.77) | 25.68 (8.14) | 29.59 (8.85) |

correlations between non-symbolic and symbolic ordering increased steadily over the study period (Grade 1: $r(155) = .54$, Grade 2: $r(155) = .63$, and Grade 3: $r(155) = .70$, all $p$s < .001). We conducted Fisher's $z$ tests [55] to analyze whether this increase in concurrent correlations was statistically significant. This revealed that the correlation between non-symbolic and symbolic order processing was higher in Grade 3 than in Grade 1 ($z = 2.29$, $p = .011$). The other concurrent correlations did not differ significantly (Grade 1 vs. Grade 2: $z = 1.10$, $p = 0.136$; Grade 2 vs. Grade 3: $z = 1.20$, $p = .116$).

### Correlational analysis

Zero-order correlations between the predictors as well as descriptive statistics are shown in Table 2. All numerical tasks (i.e., non-symbolic and symbolic magnitude comparison, as well as counting) were significantly related to processing speed, and this was expected due to the speeded nature of these tasks. Importantly, the correlation analyses showed that storage and manipulation components of verbal and visuo-spatial working memory are distinct constructs: We found only small-to-medium associations between performance in the two forward and the two backward conditions. Our results also show a close relation between non-symbolic and symbolic magnitude comparison.

Table 3 presents bivariate correlations between predictor variables and performance on the order processing tasks. There was a small significant relation between processing speed and

**Table 2. Descriptive statistics and correlations between predictors of order processing at T1.**

| Predictors |  | Processing speed | Verbal storage | Verbal manipulation | Visuo-spatial storage | Visuo-spatial manipulation | Non-symbolic comparison | Symbolic comparison | Counting |
|---|---|---|---|---|---|---|---|---|---|
| **Verbal storage** | *r* | .07 |  |  |  |  |  |  |  |
|  | *p* | .389 |  |  |  |  |  |  |  |
| **Verbal manipulation** | *r* | .06 | .25 |  |  |  |  |  |  |
|  | *p* | .494 | .002 |  |  |  |  |  |  |
| **Visuo-spatial storage** | *r* | .21 | .15 | .24 |  |  |  |  |  |
|  | *p* | .009 | .069 | .003 |  |  |  |  |  |
| **Visuo-spatial manipulation** | *r* | .15 | .17 | .27 | .42 |  |  |  |  |
|  | *p* | .057 | .038 | .001 | < .001 |  |  |  |  |
| **Non-symbolic comparison** | *r* | .38 | .09 | .11 | .24 | .36 |  |  |  |
|  | *p* | < .001 | .257 | .181 | .003 | < .001 |  |  |  |
| **Symbolic comparison** | *r* | .53 | .01 | .11 | .33 | .35 | .71 |  |  |
|  | *p* | < .001 | .895 | .184 | < .001 | < .001 | < .001 |  |  |
| **Counting** | *r* | .38 | .06 | -.04 | .10 | .13 | .24 | .29 |  |
|  | *p* | < .001 | .460 | .622 | .215 | .109 | .002 | < .001 |  |
|  | *M* | 31.47 | 25.66 | 10.16 | 22.66 | 13.66 | 59.94 | 37.16 | 1.31 |
|  | *SD* | 6.31 | 3.38 | 2.63 | 3.14 | 4.50 | 16.88 | 9.19 | 0.26 |

*Note.* All measures except counting were quantified by the sum of correct answers. Counting was quantified as N correct/second.

**Table 3. Bivariate correlations between predictors and the order processing tasks.**

| | Non-symbolic ordering | | | | | | Symbolic ordering | | | | | |
| | T1 | | T2 | | T3 | | T1 | | T2 | | T3 | |
| | r | p | r | p | r | p | r | p | r | p | r | p |
|---|---|---|---|---|---|---|---|---|---|---|---|---|
| Processing speed | .27 | .001 | .36 | < .001 | .33 | < .001 | .32 | < .001 | .33 | < .001 | .36 | < .001 |
| Verbal storage | .09 | .28 | .12 | .13 | .04 | .65 | .11 | .19 | .09 | .26 | .10 | .23 |
| Verbal manipulation | .08 | .16 | .12 | .13 | .12 | .15 | .08 | .30 | .20 | .01 | .20 | .01 |
| Visuo-spatial storage | .18 | .03 | .33 | < .001 | .30 | < .001 | .26 | .001 | .32 | < .001 | .32 | < .001 |
| Visuo-spatial manipulation | .37 | < .001 | .46 | < .001 | .26 | .001 | .44 | < .001 | .45 | < .001 | .40 | < .001 |
| Non-symbolic comparison | .50 | < .001 | .48 | < .001 | .41 | < .001 | .53 | < .001 | .34 | < .001 | .31 | < .001 |
| Symbolic comparison | .39 | < .001 | .51 | < .001 | .45 | < .001 | .46 | < .001 | .44 | < .001 | .44 | < .001 |
| Counting | .18 | .03 | .22 | .007 | .13 | .11 | .32 | < .001 | .27 | < .001 | .18 | .02 |

the speeded ordering tasks at all time points. The relation between verbal memory (storage and manipulation) and ordering tasks (non-symbolic and symbolic) was weak and mostly non-significant. There was only a small significant association between performance on the verbal manipulation task and symbolic ordering in Grades 2 and 3. In contrast, visuo-spatial memory tasks were significantly related to both order processing tasks. Whilst the correlations between visuo-spatial storage and ordering were weak, visuo-spatial manipulation was more closely related to ordering. Notably, both non-symbolic and symbolic magnitude comparison skills were significantly related to both non-symbolic and symbolic order processing. Consistent with our hypotheses, counting was significantly related to symbolic ordering. The relation between counting and non-symbolic ordering was overall weaker (and statistically not significant in Grade 3). Fisher's z test confirmed that counting was more strongly associated with symbolic than non-symbolic order processing in Grade 1 (rs .32 vs .18, z = 1.75, p = .03). For the longitudinal association of counting with ordering in Grades 2 and 3, this difference was not significant (Grade 2: rs .27 vs. .22, z = .75, p = .23; Grade 3: rs .18 vs. .13, z = .81, p = .21).

## Concurrent prediction of non-symbolic and symbolic order processing in Grade 1

To predict non-symbolic and symbolic order processing in Grade 1, we performed hierarchical linear regression analyses predicting non-symbolic and symbolic order processing in Grade 1. As can be seen in Table 4, the regression models yielded an adjusted $R^2$ of .26, $F(8,148) = 146.25$, $p < .001$ for non-symbolic ordering, and an adjusted $R^2$ of .36, $F(8,148) = 352.41$, $p < .001$ for symbolic ordering. Our control variable processing speed could explain 7% of variance in non-symbolic ordering and 10% in symbolic ordering. Introducing the storage components of working memory (digit span and block-tapping forward) only contributed a negligible amount of variance to non-symbolic ordering, and 3% of incremental variance to the prediction of symbolic order processing over and above processing speed. Adding the working memory components involving manipulation (digit span and block tapping backward) resulted in another 9% of incremental variance in non-symbolic ordering, as well as 11% of incremental variance in symbolic ordering. Notably, visuo-spatial manipulation was a significant predictor of both non-symbolic and symbolic ordering, whereas verbal manipulation was not. Magnitude comparison skills contributed 11% of unique variance to the prediction of non-symbolic order processing and 10% of variance to the prediction of symbolic ordering, over and above processing speed and working memory. While non-symbolic magnitude comparison was a significant predictor of both non-symbolic and symbolic order

**Table 4. Hierarchical linear regression analyses predicting performance in non-symbolic and symbolic ordering in Grade 1 by processing speed, working memory storage and manipulation, as well as non-symbolic and symbolic numerical skills.**

| Models | Model 1 Processing speed | | Model 2 Working memory storage | | Model 3 Working memory manipulation | | Model 4 Magnitude comparison | | Model 5 Counting | |
|---|---|---|---|---|---|---|---|---|---|---|
| | β | p | β | p | β | p | β | p | β | p |
| **Non-symbolic ordering** | | | | | | | | | | |
| Processing speed | .27 | .001 | .24 | .003 | .22 | .005 | .09 | .261 | .08 | .329 |
| Verbal storage | | | .05 | .508 | .02 | .773 | .02 | .833 | .01 | .848 |
| Visuo-spatial storage | | | .12 | .129 | -.01 | .923 | -.03 | .741 | -.03 | .745 |
| Verbal manipulation | | | | | -.03 | .715 | -.03 | .713 | -.02 | .742 |
| Visuo-spatial manipulation | | | | | .35 | < .001 | .24 | .004 | .23 | .005 |
| Non-symbolic comparison | | | | | | | .40 | < .001 | .40 | < .001 |
| Symbolic comparison | | | | | | | -.01 | .914 | -.01 | .899 |
| Counting | | | | | | | | | .03 | .712 |
| Model Fit | $F = 12.01$, $p = .001$, adjusted $R^2 = .07$ | | $F = 5.06$, $p = .002$, adjusted $R^2 = .07$ | | $F = 6.83$, $p < .001$, adjusted $R^2 = .16$ | | $F = 9.14$, $p < .001$, adjusted $R^2 = .27$ | | $F = 7.97$, $p < .001$, adjusted $R^2 = .26$ | |
| **Symbolic ordering** | | | | | | | | | | |
| Processing speed | .32 | < .001 | .28 | < .001 | .26 | < .001 | .12 | .125 | .07 | .384 |
| Verbal storage | | | .06 | .437 | .03 | .669 | .03 | .673 | .02 | .754 |
| Visuo-spatial storage | | | .19 | .015 | .05 | .530 | .03 | .721 | .03 | .698 |
| Verbal manipulation | | | | | -.06 | .451 | -.06 | .428 | -.04 | .564 |
| Visuo-spatial manipulation | | | | | .39 | < .001 | .27 | < .001 | .27 | .001 |
| Non-symbolic comparison | | | | | | | .36 | < .001 | .35 | < .001 |
| Symbolic comparison | | | | | | | .04 | .719 | .03 | .806 |
| Counting | | | | | | | | | .16 | .022 |
| Model Fit | $F = 18.16$, $p < .001$, adjusted $R^2 = .10$ | | $F = 8.58$, $p < .001$, adjusted $R^2 = .13$ | | $F = 10.70$, $p < .001$, adjusted $R^2 = .24$ | | $F = 12.61$, $p < .001$, adjusted $R^2 = .34$ | | $F = 12.03$, $p < .001$, adjusted $R^2 = .36$ | |

processing, the symbolic version of the magnitude comparison task did not contribute any incremental variance. As expected, introducing counting in the last step of the regression model did not add any variance to the prediction of non-symbolic ordering, whereas it contributed a small but significant amount of variance (2%) to the prediction of symbolic ordering. (For standard errors and confidence intervals of the regression coefficients, see supporting information, S1 Table).

### Longitudinal prediction of developmental change in order processing

We calculated linear regression analyses to predict developmental change in symbolic and non-symbolic order processing between Grades 1 and 2, and Grades 2 and 3. As can be seen in Table 5, change in non-symbolic order processing from Grade 1 to 2 was significantly predicted by visuo-spatial manipulation and symbolic magnitude processing. These two variables were also significant predictors of change in symbolic order processing from Grade 1 to 2. Unique variance in symbolic order processing was also explained by non-symbolic magnitude comparison, but note that the regression coefficient was negative. The bivariate correlation between this predictor and the outcome variable was $r = .03$. Thus, despite being uncorrelated with the outcome, non-symbolic magnitude comparison still contributed to the prediction, alluding to a classical suppression effect. Suppressor variables improve prediction of the criterion by being correlated with other predictors and suppressing criterion-irrelevant variance in these predictors [56]. In the present case, a possible explanation for the observed finding is that non-symbolic magnitude processing eliminated variance from other predictors irrelevant for the prediction of change in

**Table 5. Linear regression analyses predicting change in non-symbolic and symbolic ordering from Grade 1–2 by processing speed, working memory storage and manipulation, non-symbolic and symbolic comparison, as well as counting.**

| | Non-symbolic ordering | | | Symbolic ordering | | |
|---|---|---|---|---|---|---|
| | β | sr | p | β | sr | p |
| Processing speed | .08 | .07 | .371 | .07 | .06 | .442 |
| Verbal storage | .04 | .04 | .577 | -.02 | -.02 | .821 |
| Visuo-spatial storage | .10 | .09 | .227 | .06 | .05 | .480 |
| Verbal manipulation | -.02 | -.02 | .840 | .13 | .12 | .117 |
| Visuo-spatial manipulation | .18 | .15 | .041 | .18 | .15 | .045 |
| Non-symbolic comparison | -.03 | -.02 | .777 | -.31 | -.21 | .006 |
| Symbolic comparison | .25 | .15 | .039 | .29 | .18 | .019 |
| Counting | .02 | .02 | .839 | .04 | .04 | .642 |
| Model Fit | $F = 4.42$, $p < .001$, adjusted $R^2 = .15$ | | | $F = 13.29$, $p = .002$, adjusted $R^2 = .11$ | | |

*Note.* For all predictors, standardized regression coefficients are reported. *sr* refers to the semipartial correlation between a given predictor and ordering processing.

symbolic order processing. In other words, the significant contribution of non-symbolic magnitude processing to the prediction of change in symbolic order processing may be due to the fact that it enhanced the predictive value of other variables. (For standard errors and confidence intervals of the regression coefficients, see supporting information, S2 Table).

To test this assumption, we ran an additional hierarchical linear regression analysis predicting change in symbolic ordering from Grade 1 to 2, in which we entered all predictors in a first step, and only non-symbolic magnitude comparison in a second step (S4 Table). In this analysis, the predictive contribution of symbolic magnitude comparison was small and non-significant when non-symbolic magnitude comparison introduced to the model in the first step (regression coefficient: $B = .08$, $p = .395$). In contrast, there was a clearly higher and statistically significant unique predictive contribution of symbolic magnitude processing once non-symbolic magnitude comparison was added in the second step (regression coefficient: $B = .29$, $p = .019$).

Turning to developmental change from Grade 2 to Grade 3 (Table 6), none of our predictors predicted change in either of the order processing tasks from Grade 2 to Grade 3. (For standard errors and confidence intervals of the regression coefficients, see supporting information, S3 Table).

**Table 6. Linear regression analyses predicting change in non-symbolic and symbolic ordering from Grade 2–3 by processing speed, working memory storage and manipulation, non-symbolic and symbolic comparison, as well as counting.**

| | Non-symbolic ordering | | | Symbolic ordering | | |
|---|---|---|---|---|---|---|
| | β | sr | p | β | sr | p |
| Processing speed | .10 | .08 | .327 | .15 | .12 | .123 |
| Verbal storage | -.06 | -.05 | .508 | .02 | .02 | .777 |
| Visuo-spatial storage | .13 | .11 | .167 | .06 | .05 | .505 |
| Verbal manipulation | -.09 | -.08 | .634 | .04 | .03 | .792 |
| Visuo-spatial manipulation | .04 | .04 | .320 | .02 | .02 | .686 |
| Non-symbolic comparison | .12 | .08 | .317 | -.06 | -.04 | .645 |
| Symbolic comparison | .09 | .06 | .459 | .13 | .08 | .331 |
| Counting | -.06 | -.06 | .456 | -.10 | -.09 | .245 |
| Model Fit | $F = 1.59$, $p = .131$, adjusted $R^2 = .03$ | | | $F = 1.19$, $p = .310$, adjusted $R^2 = .01$ | | |

*Note.* For all predictors, standardized regression coefficients are reported. *sr* refers to the semipartial correlation between a given predictor and ordering processing.

## Discussion

The present study aimed to advance our understanding of the cognitive foundations of order processing in three critical ways: First, we wanted to investigate the common and distinct numerical mechanisms supporting non-symbolic and symbolic order processing at the beginning of primary school. Our second goal was to disentangle the influence of storage and manipulation components of verbal and visuo-spatial working memory on order processing, and thirdly, we wanted to explore the cognitive determinants of developmental change in order processing. This is the first longitudinal investigation of the cognitive predictors of order processing and its development over a three-year period. We employed a novel and reliable paper-and-pencil measure of ordinal processing. Children's task performance in both the non-symbolic and symbolic ordering tasks increased across the grades, indicating that this measure is well-suited to capture the longitudinal development of ordinal processing. Our study contributes to the previous literature by showing a steadily increasing association between non-symbolic and symbolic ordering over the course of Grades 1–3, suggesting that numerical order processing may develop towards one unitary factor.

### Common and distinct numerical predictors of non-symbolic and symbolic order processing

Concerning the first aim of our study, we found that non-symbolic and symbolic order processing are supported by predominantly similar cognitive mechanisms, with the notable exception of a small unique contribution of counting to symbolic order processing.

Irrespective of the number format of the ordering task, the ability to perform non-symbolic number comparisons significantly predicted children's order processing skills in Grade 1. We observed a substantial concurrent correlation between non-symbolic magnitude comparison and both non-symbolic and symbolic ordering in Grade 1 ($r = .50 - .53$), and this predictive relation was significant even after accounting for processing speed and working memory. This is especially noteworthy given that previous meta-analyses [57–59] identified only a small correlation between non-symbolic number comparisons and higher-order mathematical abilities ($r = .20 - .24$). The observed association between cardinal and ordinal processing supports the notion that order judgements rely to some extent on multiple pair-wise magnitude comparisons [14]. Critically, our results extend previous theoretical conceptions by showing that pair-wise magnitude comparisons are not an exclusive feature of non-symbolic ordering: At least in our young sample, non-symbolic magnitude comparison also served as a unique predictor of symbolic ordering. Our findings thus suggest that non-symbolic magnitude processing plays a significant role when dealing with ordered sequences of numbers at the beginning of primary school. This is interesting, given that already infants appear to be able to infer the order of non-symbolic sequences [60], while symbolic ordering skills only emerge later in development [61]. It is plausible that symbolic order processing may partly depend on approximation at the very beginning of primary school, because children are still acquiring proficiency with the symbolic number system during this period [27,46]. Once children have gained sufficient familiarity with symbolic numbers, other mechanism may play an increasingly prominent role.

The relatively strong correlation between non-symbolic magnitude comparison and order processing found in this study might be partly due to similarities between the measures. One common feature is the timed paper-and-pencil task format (i.e., we measured the number of correct judgements within a limited time span to assess children's efficiency). However, non-symbolic magnitude processing predicted order processing even when we controlled for processing speed, which was assessed in a very similar (but non-numerical) task format. This

finding indicates that the relation between magnitude and order processing is not fully explained by the shared influence of processing speed or task format.

A critical question of this study was whether we would observe unique contributions of symbolic numerical skills to symbolic, but not non-symbolic order processing. Our findings showed that symbolic magnitude comparison was correlated with symbolic as well as non-symbolic order processing, but did not explain unique variance above and beyond non-symbolic magnitude processing. Counting, however, was selectively relevant for symbolic order processing, even though its contribution was relatively minor.

As previously pinpointed [15], the relation between symbolic cardinal and ordinal processing is subject to dynamic changes at the beginning of formal instruction. The fact that no unique variance was accounted for by symbolic magnitude processing in Grade 1 corroborates cross-sectional results [15], suggesting that the contribution of symbolic magnitude processing to ordering has not yet emerged in Grade 1. However, note that non-symbolic and symbolic magnitude processing performance were strongly correlated in our study, calling into question whether both really measure distinct constructs. Indeed, some researchers [47,62] have proposed a unitary factor underlying non-symbolic and symbolic magnitude comparison skills in school-age children.

As expected, we found that symbolic order processing draws on cognitive mechanisms extending beyond pairwise magnitude comparisons. Specifically, children's counting skills uniquely predicted variance in symbolic order processing skills, over and above processing speed, working memory and cardinal magnitude processing. In other words, being able to correctly retrieve the order of verbal number words in the counting sequence poses an advantage when performing symbolic order judgements. Our findings support the assumption that retrieval from the verbal count list stored in long-term memory provides a mechanism distinguishing symbolic ordering from its non-symbolic counterpart [14]. This retrieval-based mechanism provides a plausible explanation for the existence of reversed distance effects in symbolic ordering, which are absent from non-symbolic ordering [5,8]. It may also at least partly explain why children were consistently able to solve more symbolic ordering items within the given time limit at all timepoints.

Note that we employed a different counting task than an earlier study, which did not report a similar relation between performance in a dot enumeration task and symbolic order processing [20]. Arguably, by asking children to verbally recite the count list, we may have used a more direct measure of familiarity with the count list. Further studies addressing this explanation and replicating our findings are clearly needed. However, our results suggest that the role of counting for symbolic order processing should not be discounted.

A major finding of the present study is that visuo-spatial manipulation contributed to both non-symbolic and symbolic ordering. This clearly shows that the ability to manipulate task-relevant information throughout multiple processing steps is involved in order processing [33–35]. Together with the observation that ordering tasks are solved at least partly by pairwise magnitude comparison, it is possible that visuo-spatial manipulation may be involved in splitting ordered triplets into two sequential magnitude comparison tasks. This is important because many studies did not consider the contribution of visuo-spatial working memory to order processing, instead focusing on serial-order working memory [43] or verbal working memory [2,36,37], but see also [4]. Our study supports the proposition that visuo-spatial memory [4,41] may be an important prerequisite for learning the spatial-ordinal relation of digits. Notably, our findings extend previous research by suggesting that this is also true for ordered sequences of non-symbolic numerosities. However, it is important to discuss the possibility that the observed predictive contribution of visuo-spatial working memory to order processing may be specific to younger children. Indeed, literature on arithmetic development suggests an

especially prominent contribution of visuo-spatial working memory in children at the very beginning of formal mathematics instruction [42].

An important aspect of our study design was that we used specific measures to assess storage and manipulation components of working memory. The present results showed that visuo-spatial storage did not explain unique variance in non-symbolic order processing, and its contribution to symbolic order processing was fairly minor. This is contrasted by a prominent unique contribution of visuo-spatial manipulation, explaining an additional 9% of variance in non-symbolic and 11% of variance in symbolic ordering. This evidence strongly suggests that being able to manipulate visuo-spatial information is more important for order processing than storing information over short periods of time.

In contrast to our expectations, no associations were observed between verbal storage and manipulation and non-symbolic and symbolic order processing. This is surprising, given that previous studies reported significant associations between verbal working memory and ordering in children [36] and adults [37]. However, as noted above, it is well-established that the contribution of different working memory components to arithmetic development changes as a function of age [42]. Thus, it is entirely possible that verbal working memory may become more important for order processing at a later point in development when children acquire a better understanding of ordered sequences. Indeed, our current results showed a significant longitudinal correlation between verbal working memory in Grade 1 and symbolic order processing in Grades 2 and 3, whereas the concurrent correlation at the beginning of the study period in Grade 1 was small and non-significant.

Moreover, note that previous studies on the relation between working memory and order processing employed complex working memory span tasks. These are highly demanding tasks not only requiring individuals to store and rehearse information, but also to simultaneously process additional information (e.g., answering questions whilst remembering certain words across several questions). Such working memory span paradigms thus require considering new stimuli which are interfering with the primary storage task while backward digit span is limited to carrying out mental transformations [63]. Future research will have to determine the causal mechanisms explaining why being able to deal with interfering verbal information is especially relevant for order processing, over and above the ability to manipulate verbal information.

## Predictors of developmental change

Our longitudinal design allowed us to go beyond earlier cross-sectional work by investigating the cognitive predictors of developmental change in order processing across the early school years. Results show that developmental change in both non-symbolic and symbolic ordering between Grades 1 and 2 was predicted by the manipulation component of visuo-spatial working memory and symbolic magnitude comparison. First, this supports the notion of a continuing impact of visuo-spatial manipulation on order processing and its development, irrespective of the number format. Notably, visuo-spatial manipulation was a significant predictor of both non-symbolic and symbolic ordering already at the beginning of our study period in Grade 1. Thus, good visuo-spatial manipulation skills not only enable children to apply helpful visual strategies for their order judgements, for instance by splitting a triplet of quantities or digits into two consecutive magnitude comparison items. They also help them to improve their order processing skills across the following school year. Future studies should aim to unravel whether visuo-spatial skills were particularly important in the present study due to the visual nature of our order processing tasks. An interesting avenue for future research would be to investigate the predictive contribution of visuo-spatial manipulation to performance in an auditory ordering task in which participants are presented with triplets of

verbal number words or sequences of sounds. In light of the converging evidence [2–5] pointing towards a strong association between symbolic order processing and arithmetic performance it would also be interesting to explore whether the visual component of order processing makes this task especially relevant for arithmetic.

Symbolic magnitude processing did not explain a significant amount of variance in children´s order processing performance concurrently, in Grade 1. Nevertheless, it emerged as a longitudinal predictor of developmental change in non-symbolic as well as symbolic order processing between Grades 1 and 2. The predictive contribution of symbolic magnitude comparison to non-symbolic ordering was somewhat surprising, given that we had expected a particularly strong predictive contribution of non-symbolic magnitude processing skills to non-symbolic order processing. This finding thus cannot corroborate the assumption of distinct neurocognitive developmental trajectories for non-symbolic and symbolic number skills [23,44,45], but instead points towards an influence of symbolic number skills on the development of non-symbolic number skills at the beginning of primary school [64]. However, it is again important to interpret this finding with care given the strong correlation between the two magnitude comparison conditions. This is also relevant for the suppression effect of non-symbolic magnitude comparison we observed: It increased the predictive power of symbolic magnitude comparison by eliminating irrelevant variance in this measure for the prediction of symbolic order processing. This means that there is a common facet of non-symbolic and symbolic magnitude comparison that is in fact irrelevant for the development of symbolic ordering skills. Arguably, this shared facet is the common task format: Including non-symbolic magnitude comparison in the prediction of change in order processing probably improves the prediction of symbolic magnitude comparison by eliminating the portion of variance due to pairwise magnitude comparisons. A plausible interpretation of this pattern is that performing pairwise magnitude comparisons may not be essential for the development of order processing after Grade 1. Instead, children's familiarity with symbolic numbers *per se* may be more critical for acquiring proficiency in symbolic order processing than their ability to compare sets of numbers. This line of reasoning corresponds with recent evidence suggesting that children's knowledge of symbolic numbers is the most important predictor of arithmetic at the beginning of primary school [62].

Among the predictors considered in the current study, none could uniquely explain developmental change in order processing from Grade 2 to 3. There are several explanations to consider: First, this finding may partly be due to the high stability of interindividual differences in order processing during this time window ($r = .71$). Second, our predictors were assessed at the beginning of the study in Grade 1. Given the only moderate stability of our predictors in this age group, the small and non-significant effect sizes we observed are not entirely surprising. Nonetheless, we acknowledge that children improved their ordering performance between Grade 2 and 3, and we could not identify the variables driving this development. In this period, children are expected to acquire a readily accessible storage of arithmetic facts in long-term memory and become fluent in mental calculations. This raises the possibility that the acquisition of arithmetic competencies may foster the development of number ordering. In other words, not only order processing may promote arithmetic development [3,15,43], but also vice-versa. Investigating the possibly bi-directional relation between order processing and arithmetic provides an exciting avenue for future research.

## Limitations

Our ordinal processing tasks only included items in ascending order. We piloted task versions with a mix of ascending and descending items but realized that such a task version was too difficult for children in Grade 1.

It is important to acknowledge that our paper-and-pencil task format did not allow us to calculate reversed distance effects, which are a widely used measure of order processing. However, previous evidence suggested that reversed distance effects may not be an entirely unproblematic measure of order processing in young samples: While significant reverse distance effects were reported as early as in Grade 1 at the group level [20], a different picture emerged when reaction time patterns of first and second graders were analyzed at the individual level [15]: More than half of children did not show the expected reversed distance effect at all.

Lastly, as the ordering tasks were administered in a classroom setting, we cannot fully rule out that some children employed counting strategies despite our explicit instruction not to count the dot arrays in the non-symbolic ordering tasks. Thus, in some cases children may have retrieved the verbal labels associated with the non-symbolic quantities. However, note that we did not find any significant association between children's verbal counting skills and their non-symbolic ordering performance. Nonetheless, to minimize the possibility of verbal enumeration in non-symbolic ordering tasks, researchers may employ computerized tasks in which stimuli are only briefly presented for a limited time span (e.g., 500 ms). Moreover, it is a well-established fact that non-numerical continuous dimensions such as size or density play a role in non-symbolic magnitude comparison tasks [65]. Thus, future research should address whether non-symbolic numerical judgements are similarly biased by non-numerical cues.

## Conclusion

Our study shows that non-symbolic and symbolic order processing rely on largely similar, but partly distinct cognitive mechanisms. While both order processing tasks were similarly predicted by magnitude processing and visuo-spatial working memory, only initial performance in symbolic ordering was additionally supported by retrieval from the verbal count-list. These findings advance our theoretical understanding of the cognitive foundations of order processing by indicating that symbolic ordering engages cognitive mechanisms extending beyond pairwise magnitude comparisons and working memory.

## Supporting information

**S1 Table. Hierarchical linear regression analyses predicting performance in non-symbolic and symbolic ordering in Grade 1 by processing speed, working memory storage and manipulation, as well as non-symbolic and symbolic numerical skills.** Note. For all predictors, unstandardized regression coefficients are reported (standard errors in parentheses). LCI denotes the lower end of the 95% confidence interval, and UCI denotes the upper end of the 95% confidence interval.
(DOCX)

**S2 Table. Linear regression analyses predicting change in non-symbolic and symbolic ordering from Grade 1–2 by processing speed, working memory storage and manipulation, non-symbolic and symbolic comparison, as well as counting.** Note. For all predictors, unstandardized regression coefficients are reported (standard errors in parentheses). LCI denotes the lower end of the 95% confidence interval, and UCI denotes the upper end of the 95% confidence interval.
(DOCX)

**S3 Table. Linear regression analyses predicting change in non-symbolic and symbolic ordering from Grade 2–3 by processing speed, working memory storage and manipulation, non-symbolic and symbolic comparison, as well as counting.** Note. For all predictors, unstandardized regression coefficients are reported (standard errors in parentheses). LCI

denotes the lower end of the 95% confidence interval, and UCI denotes the upper end of the 95% confidence interval.
(DOCX)

**S4 Table. Hierarchical linear regression analysis predicting symbolic ordering in Grade 1, with non-symbolic magnitude comparison entered in the second step.** Note. For all predictors, unstandardized regression coefficients are reported (standard errors in parentheses). *r* refers to the correlation between a given predictor and ordering processing.
(DOCX)

## Author Contributions

**Conceptualization:** Silke M. Göbel, Karin Landerl.

**Data curation:** Sabrina Finke, Chiara Banfi, Anna F. Steiner.

**Formal analysis:** Sabrina Finke, H. Harald Freudenthaler.

**Funding acquisition:** Silke M. Göbel, Karin Landerl.

**Investigation:** Sabrina Finke, Chiara Banfi, Anna F. Steiner.

**Methodology:** Stephan E. Vogel, Silke M. Göbel, Karin Landerl.

**Project administration:** Silke M. Göbel, Karin Landerl.

**Resources:** Silke M. Göbel, Karin Landerl.

**Supervision:** Silke M. Göbel, Karin Landerl.

**Visualization:** Sabrina Finke.

**Writing – original draft:** Sabrina Finke, Chiara Banfi, Karin Landerl.

**Writing – review & editing:** Sabrina Finke, Chiara Banfi, H. Harald Freudenthaler, Anna F. Steiner, Stephan E. Vogel, Silke M. Göbel, Karin Landerl.

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
