## [Decision Letter · Decision Letter 0]

14 Jul 2021

PONE-D-21-14700

Common and distinct predictors of non-symbolic and symbolic ordinal number processing across the early primary school years

PLOS ONE

Dear Dr. Finke,

Thank you for submitting your manuscript to PLOS ONE. I have sent it to two expert reviewers and have now received their comments. As you will see at the bottom of this email, both reviewers find that your manuscript is well written, sound, and addresses an important question. I concur with the reviewers. However, you will see that both reviewers suggest changes to improve the overall quality of the paper. As the reviewers have been very clear in their suggestions for improvement I will not repeat them here. However, I would encourage you to address these points in a minor revision. 

We look forward to receiving your revised manuscript.

Kind regards,

Jérôme Prado

Academic Editor

PLOS ONE

Journal Requirements:

4.  Thank you for stating the following in the  Financial Disclosure  section:

"This work was supported by the Austrian Science Fund (www.fwf.at), grant number I

2778-G16 awarded to KL, and the Economic and Social Research Council

(www.esrc.ukri.org), grant number ES/N014677/1 awarded to SMG . The funders had

no role in study design, data collection and analysis, decision to publish, or preparation

of the manuscript."

We note that one or more of the authors are employed by a commercial company: "BioTechMed-Graz"

Reviewers' comments:

Reviewer #1: This manuscript is about a longitudinal study, involving 157 children from Grade 1 to 3, investigating the contribution of different variables to performance and developmental change in non-symbolic and symbolic numerical order processing.

Numerical ordering seems to be the forgotten process in the field of numerical cognition. There are only few contributions in this precise field, which have however, pinpointed the importance of ordering abilities for developing good early arithmetical skills in children.

Accordingly, this manuscript deal with an interesting subject and shed light on how these particular skills might develop through early formal schooling.

I think this is a contribution that opens up future research.

In general, I found the methodology and results part to be clear and understandable.

Regarding the introduction, and to a lesser extent the discussion, I feel that it is necessary for the reader to have in-depth knowledge of the field to be able to follow the reasoning of the authors. Without knowledge of the cited papers, it seems complicated to me to be able to understand the underling logic.

This is why I think that certain paragraphs would benefit from being slightly more explicit.

For example:

- Between line 86 and 89, it seems interesting to me to clarify why this is so and if this idea is based on elements of the literature. Moreover, I think it might be interesting for a good understanding of the reader to slightly develop this idea that "(…) has led researchers to propose that non-symbolic order processing may be closely related to cardinal processing of non-symbolic magnitudes, whereas additional retrieval-based mechanisms seem to be engaged in symbolic ordering”.

- Concerning the fMRI studies (L100) it could be interesting to specify which cerebral regions are involved and the range of the numbers used in Lyons & Beilock’ studies.

Concerning the passage of "Cardinal magnitude processing", the authors might like to consider to reiterate “symbolic” or “digit” in front of “numerical magnitude”, because it might lead to confusion (especially the first-time mentioned L 117).

L156 I I would qualify this (…) of children in Grades 1 to 6. However, enumerating dot numbers between 1 and 9 is a complex task (…).

Because it might be difficult for youngest but not for older ones.

Divers : I would have liked to have more qualitative details about how the children behave towards the task, especially the non-symbolic ordering one. As the authors mentioned that children do less of these exercises, don't they enumerate/count dots (especially younger ones) even if they were explicitly instructed not to ? In this case, children would also retrieve verbal number words but through the enumeration of the counting sequence while in the symbolic task they retrieve a verbal number word regarding a specific Arabic digit, i.e., more directly.

I would like to know the opinion of the authors on this point (and maybe consider this as a limitation of their study?). Moreover, I wonder if it would be possible and/or interesting to propose a different non-symbolic ordering task as ordering continuous magnitude (3 different size rectangles for example) ?

Finally, I was happy to find an explanation in the limitations section that "descending" sequences were not included. The authors might want to consider mentioning it already in the methodology part of the manuscript.

Minor comments

L216 to 223, authors are now mentioned (considered to adjust, for example line 198 if this is possible).

Define r and R2 as it is done for z (e.g., L438)

L442 p = p =

Reviewer #2: Summary

The current study examined the predictors of non-symbolic and symbolic order processing in the first years of primary school. In a longitudinal study, the authors tested whether non-symbolic and symbolic-magnitude comparison, counting and working memory were predictive for respectively non-symbolic and symbolic order processing. The results demonstrated that non-symbolic magnitude comparison and visuo-spatial manipulation were significant predictors in both non-symbolic and symbolic order processing. Counting was also predictive for symbolic order processing. Based on their findings, the authors concluded that there is partial overlap of the cognitive mechanisms that underlie non-symbolic and symbolic order processing.

Evaluation

The findings of the manuscript are highly contributive as they shed light on the performance and developmental change in symbolic and non-symbolic order processing. The manuscript is well-written and the findings are clearly reported. I also appreciated that the data is made available on the Open Science Framework. Overall, I think the manuscript makes an important contribution and is suitable for publication in PLOS ONE. Below I outline a few questions and suggestions to think about while revising the manuscript.

Major comments

1. Sensitivity analsis

It is reported that a sample of 157 is needed to detect an incremental effect of ΔR2 = .05, but the expected alpha and power is not mentioned. Could it be clarified which alpha and power parameters were entered in the sensitivity analysis?

2. Presentation order of the task

What was the order in which the tasks were presented at each time point? Did all the children perform the task in the same order or was the presentation order of tasks counterbalanced? This could be clarified in the procedure.

3. Items on the order tasks

With regard to the non-symbolic order processing and symbolic order processing task, it is mentioned that there were 41 ascending triplets for the non-symbolic task and 35 ascending triplets for the symbolic task among 80 items. Were the remaining trials all non-ordered? In the last paragraph about the limitations of the study, it is mentioned that no descending trials were excluded, so I assume that only ascending and non-ordered trials were included but this could be clarified in the method section. Furthermore, I wondered why the authors did not choose to present half of the items in ascending order since an uneven distribution of items in the different conditions could have led to a response bias.

4. Mean performance on the tasks

It would be informative to not only report the descriptive statistics for the order tasks, but also for the other tasks such as the working memory tasks and counting task. In addition, I wondered whether there were any outliers for the symbolic order processing task, working memory tasks and counting task. Outliers have been reported for the non-symbolic order processing task but not for the other tasks.

5. The predictive role of verbal and visual-spatial working memory

Could the finding that visuospatial working memory was a strong predictor for order processing be specific for younger children? Literature about the development of arithmetic performance shows that visual-spatial working memory is especially a strong predictor when children are not familiar yet with the arithmetic operations. When arithmetic operations become more familiar, verbal working memory plays a more important role (see Ragubar et al., 2010 for a review). A similar process could play a role in the development of order processing and verbal working memory could become more important for order processing as children acquire a better understanding of ordered sequences. In line with this reasoning, the authors do find a significant correlation between verbal manipulation and symbolic ordering for time point 2 and 3 while they did not observe this correlation for time point 1.

References

Raghubar, K. P., Barnes, M. A., & Hecht, S. A. (2010). Working memory and mathematics: A review of developmental, individual difference, and cognitive approaches. Learning and Individual Differences,20(2), 110-122. https://doi.org/10.1016/j.lindif.2009.10.005

---

## [Author Response · Author response to Decision Letter 0]

11 Aug 2021

Academic Editor comments

Q(1): Please ensure that your manuscript meets PLOS ONE's style requirements, including those for file naming.

*Response: We have ensured that our manuscript meets all style requirements, including those for file naming.

Q(2): Please review your reference list to ensure that it is complete and correct. If you have cited papers that have been retracted, please include the rationale for doing so in the manuscript text, or remove these references and replace them with relevant current references. Any changes to the reference list should be mentioned in the rebuttal letter that accompanies your revised manuscript. If you need to cite a retracted article, indicate the article’s retracted status in the References list and also include a citation and full reference for the retraction notice.

*Response: The exact order of the references in the Reference list changed following the revisions made, as two papers were already cited at an earlier point in the manuscript. Also, we have added another reference to the References list following a reviewer’s suggestion:

65. Leibovich T, Ansari D. The symbol-grounding problem in numerical cognition: A review of theory, evidence, and outstanding questions. Can J Exp Psychol. 2016;70(1):12–23. doi: 10.1037/cep0000070.

Moreover, we noticed that one reference was previoulusy listed twice in the reference list. We thus deleted the second occurrence:

62. Sasanguie D, Vos H. About why there is a shift from cardinal to ordinal processing in the association with arithmetic between first and second grade. Dev Sci. 2018 Sep;21(5):e12653. doi: 10.1111/desc.12653. 

We also double-checked our in-text citations. We noticed two mistakes in our in-text citations:

1.) line 56: “A growing body of evidence has revealed the unique predictive contribution of ordinal processing skills to arithmetic performance in children and adults (e.g., (2–6))“ instead of (e.g., (1-5))

2.) line 376-377: “(…) Working Memory Test Battery for Children (WMTB-C, (50); test-retest reliability is .63, (51))“ instead of “(…) Working Memory Test Battery for Children (WMTB-C, (43); test-retest reliability is .63, (50))“.

Q(3): We note that you have stated that you will provide repository information for your data at acceptance. Should your manuscript be accepted for publication, we will hold it until you provide the relevant accession numbers or DOIs necessary to access your data. If you wish to make changes to your Data Availability statement, please describe these changes in your cover letter and we will update your Data Availability statement to reflect the information you provide.

*Response: Thank you for this information. We have now made our entire data publicly available. Hence, we would like to update our Data Availability statement to reflect this decision:

“The data collected for this study are available at the UK Data Service (doi: 10.5255/UKDA-SN-854335). Data of test-retest reliability for the ordinal processing tasks are publicly available on the Open Science Framework at https://osf.io/gm498/ (doi: 10.17605/OSF.IO/GM498).”

Q(4): Financial Disclosure: We note that one or more of the authors are employed by a commercial company: "BioTechMed-Graz" (…) If your commercial affiliation did play a role in your study, please state and explain this role within your updated Funding Statement.

*Response: We are happy to clarify this matter. We would like to re-confirm that all co-authors do not receive any financial support from a commercial company, and have no competing interests to declare. BioTechMed is an interdisciplinary non-profit research alliance of three universities in Graz, Austria. More information can be found at https://biotechmedgraz.at/en/about/. None of the authors is supported in the form of salaries, funded projects or any other kind of financial remuneration from BioTechMed. Therefore, we have not made any changes to our Financial Disclosure, Funding or Competing Interests Statements.

Reviewer #1 comments

Reviewer #1: This manuscript is about a longitudinal study, involving 157 children from Grade 1 to 3, investigating the contribution of different variables to performance and developmental change in non-symbolic and symbolic numerical order processing.

Numerical ordering seems to be the forgotten process in the field of numerical cognition. There are only few contributions in this precise field, which have however, pinpointed the importance of ordering abilities for developing good early arithmetical skills in children.

Accordingly, this manuscript deal with an interesting subject and shed light on how these particular skills might develop through early formal schooling.

I think this is a contribution that opens up future research.

In general, I found the methodology and results part to be clear and understandable.

Regarding the introduction, and to a lesser extent the discussion, I feel that it is necessary for the reader to have in-depth knowledge of the field to be able to follow the reasoning of the authors. Without knowledge of the cited papers, it seems complicated to me to be able to understand the underling logic.

This is why I think that certain paragraphs would benefit from being slightly more explicit.

*Response: We thank the reviewer for the overall positive evaluation of our study. We are grateful for the reviewer´s feedback about the readability of our paper and detail below, how we integrated the specific comments in our revision.

Q(1): Between line 86 and 89, it seems interesting to me to clarify why this is so and if this idea is based on elements of the literature. Moreover, I think it might be interesting for a good understanding of the reader to slightly develop this idea that "(…) has led researchers to propose that non-symbolic order processing may be closely related to cardinal processing of non-symbolic magnitudes, whereas additional retrieval-based mechanisms seem to be engaged in symbolic ordering”.

*Response Q(1): Thank you for your valuable question whether a strong link between non-symbolic order processing and pair-wise magnitude comparison was previously suggested in the literature. Indeed, this idea was previously put forward by some researchers (Lyons & Beilock, 2013; Lyons et al., 2016). We now cite these references in line 61, and extended our reasoning behind the idea that symbolic ordering may rely on retrieval-based mechanisms (lines 90-93), also adding an illustrative example. The change in lines 90-93 now reads:

“In contrast, the existence of reverse distance effects for directly ascending symbolic items may be caused by direct retrieval from the verbal count list (e.g., “1-2” is part of the counting sequence “one-two”), pointing towards the engagement of retrieval-based mechanisms in symbolic order processing (4,7,13).“

Q(2): Concerning the fMRI studies (L100) it could be interesting to specify which cerebral regions are involved and the range of the numbers used in Lyons & Beilock’ studies.

*Response Q(2): We are happy to elaborate more on the studies by Lyons and Beilock (2013, 2018). Thus, we added further information (lines 97-126):

“One fMRI study (8) examined neural signatures of cardinal and ordinal processing of symbolic numbers (visual-Arabic digits) and non-symbolic quantities (dots), as well as a non-numerical control condition (luminance). For all numerical tasks, stimuli consisted of numerosities ranging from 1 to 9. The non-symbolic and symbolic cardinal processing tasks consisted of magnitude comparison tasks with dots and digits, respectively. In both non-symbolic and symbolic ordinal processing tasks, participants were required to judge whether triplets of stimuli were ordered from left to right (increasing or decreasing) or not in order. Numerical processing was determined by subtracting brain activation associated with the control condition from each numerical task. There was a strong overlap of the neural networks involved in cardinal and ordinal processing of non-symbolic quantities: Both cardinal and ordinal judgements were associated with activations of a right-lateralized frontoparietal network, including the dorsolateral prefrontal cortex, the intraparietal sulcus and the anterior cingulate cortex. In contrast, the authors could not find any overlap in the brain activation for ordinal and cardinal processing of symbolic quantities. These results suggest a tight link between cardinal and ordinal processing of non-symbolic numbers, while such a link is less obvious for symbolic numbers. Of note, symbolic ordinal processing was selectively associated with activation of premotor regions in the left hemisphere.

A recent study (16) investigated the similarity between patterns of neural activation evoked by non-symbolic and symbolic quantities using a delayed match-to-sample task in adults. Similar to (11), stimuli consisted of numerosities between 1 and 9. Participants were required to indicate whether pairs of dot arrays or digits presented with a jittered delay were identical or non-identical. The authors considered brain activity during the presentation of the first stimulus, as well as during the delay before the onset of the second stimulus. Representational similarity analysis was conducted to compare brain activation patterns for non-symbolic and symbolic quantities. Results showed different activation patterns between non-symbolic and symbolic quantities in regions of the parietal, the frontal and occipital cortex. The activation of non-symbolic quantities was best explained by numerical ratio, whereas brain activation in response to symbolic quantities were best explained by lexical frequency. These differential patterns suggest that processing of non-symbolic and symbolic numbers is supported by distinct and largely independent cognitive mechanisms.”

Q(3): Concerning the passage of "Cardinal magnitude processing", the authors might like to consider to reiterate “symbolic” or “digit” in front of “numerical magnitude”, because it might lead to confusion (especially the first-time mentioned L 117).

*Response Q(3): Thank you for this suggestion. We accordingly clarified this when first mentioned, and also decided to reiterate this at a later point.

- Lines 140-142: Cross-sectional evidence indicates that the interplay between the ability to process the order and the magnitude of digits changes as a function of age (…)

- Line 144: symbolic order processing

Q(4): L156 I I would qualify this (…) of children in Grades 1 to 6. However, enumerating dot numbers between 1 and 9 is a complex task (…).Because it might be difficult for youngest but not for older ones.

*Response Q(4): We apologize for not being clear: by “complex task” we did not mean difficult. We are convinced that 1st graders are well able to enumerate such a low dot number. It is a complex counting task in terms of cognitive requirements as it does not only require knowledge of the count list, but also efficient application of the one-to-one principle of counting. The text was changed accordingly:

“However, enumerating dot numbers between 1 and 9 is a complex task in cognitive terms as it involves not only knowledge of the count list, but also efficient application of the one-to-one principle of counting” (line 179-181)

Q(5): I would have liked to have more qualitative details about how the children behave towards the task, especially the non-symbolic ordering one. As the authors mentioned that children do less of these exercises, don't they enumerate/count dots (especially younger ones) even if they were explicitly instructed not to ? In this case, children would also retrieve verbal number words but through the enumeration of the counting sequence while in the symbolic task they retrieve a verbal number word regarding a specific Arabic digit, i.e., more directly.

I would like to know the opinion of the authors on this point (and maybe consider this as a limitation of their study?). Moreover, I wonder if it would be possible and/or interesting to propose a different non-symbolic ordering task as ordering continuous magnitude (3 different size rectangles for example) ?

*Response Q(5): Due to the fact that the ordering tasks were administered in a classroom setting, we cannot fully exclude the possibility that children sometimes tried to enumerate or even count out the dots despite our instructions. If such attempts were observed, an experimenter would remind the child to not count. We agree with the reviewer that this should be mentioned in the limitation section. Also, we propose how future research could minimize the impact of counting strategies, and also propose that the impact of non-numerical continuous dimensions on non-symbolic ordering should be addressed (lines 778-788):

“Lastly, as the ordering tasks were administered in a classroom setting, we cannot fully rule out that some children may have employed counting strategies despite our explicit instruction not to count the dot arrays in the non-symbolic ordering tasks. Thus, in some cases children may have retrieved the verbal labels associated with the non-symbolic quantities. However, note that we did not find any significant association between children’s verbal counting skills and their non-symbolic ordering performance. Nonetheless, to minimize the possibility of verbal enumeration in non-symbolic ordering tasks, researchers may employ computerized tasks in which stimuli are only briefly presented for a limited time span (e.g., 500 ms). Moreover, it is a well-established fact that non-numerical continuous dimensions such as size or density play a role in non-symbolic magnitude comparison tasks (65). Thus, future research should address whether non-symbolic numerical judgements are similarly biased by non-numerical cues.”

Q(6): Finally, I was happy to find an explanation in the limitations section that "descending" sequences were not included. The authors might want to consider mentioning it already in the methodology part of the manuscript.

*Response Q(6): As suggested, we added the following sentence to the task description in the methodology:

“Descending items were not included, as piloting revealed that such a task version was too difficult for children in Grade 1.“ (lines 323-324)

Q(7): L216 to 223, authors are now mentioned (considered to adjust, for example line 198 if this is possible).

*Response Q(7): Thank you for noticing this! We changed this in the mentioned paragraph, and also in other relevant places:

- line 239-240: “One study reported an ordinal Stroop paradigm (…) (4) ” instead of “Vogel et al. (4) developed an ordinal Stroop paradigm (…)“

- lines 72-73: “For symbolic ordering, a reversed distance effect was reported (9), (…)” instead of “For symbolic ordering, Turconi et al. (9) reported a reversed distance effect, (…)“

- line 97: “One fMRI study (8) examined the neural signatures (…)” instead of “Lyons and Beilock (8) examined neural signatures (…)”

- lines 177-179: “A cross-sectional study applying a dot enumeration task did not find the expected association did not find the expected association between counting skills and symbolic order processing in a sample of children in Grades 1 to 6 (20)” instead of “Based on a dot enumeration task, Lyons and Ansari (20) did not find the expected association (…)“

- line 245-246: “In a similar vein, others proposed that (…) (41)” instead of “In a similar vein, Sella et al. (41) proposed that (…)“

- line 262-263: “As pinpointed previously (38), (…)” instead of “As pinpointed by Clearman et al. (38), (…)“

- line 656-658: “Note that we employed a different counting task than an earlier study (20)” instead of “Note that we employed a different counting task than an earlier study (Lyons & Ansari, 2015)”

Q(8): Define r and R2 as it is done for z (e.g., L438) 

L442 p = p =

*Response Q(8): As suggested, we have changed this paragraph (lines 459-468) and agree with the reviewer that this improves readability.

Reviewer #2 comments

Reviewer #2: Summary

The current study examined the predictors of non-symbolic and symbolic order processing in the first years of primary school. In a longitudinal study, the authors tested whether non-symbolic and symbolic-magnitude comparison, counting and working memory were predictive for respectively non-symbolic and symbolic order processing. The results demonstrated that non-symbolic magnitude comparison and visuo-spatial manipulation were significant predictors in both non-symbolic and symbolic order processing. Counting was also predictive for symbolic order processing. Based on their findings, the authors concluded that there is partial overlap of the cognitive mechanisms that underlie non-symbolic and symbolic order processing.

Evaluation

The findings of the manuscript are highly contributive as they shed light on the performance and developmental change in symbolic and non-symbolic order processing. The manuscript is well-written and the findings are clearly reported. I also appreciated that the data is made available on the Open Science Framework. Overall, I think the manuscript makes an important contribution and is suitable for publication in PLOS ONE. Below I outline a few questions and suggestions to think about while revising the manuscript.

*Response: Again, we thank this reviewer for the positive evaluation and for the detailed comments that helped us to improve the readability of our manuscript.

Q(1): Sensitivity analysis

It is reported that a sample of 157 is needed to detect an incremental effect of ΔR2 = .05, but the expected alpha and power is not mentioned. Could it be clarified which alpha and power parameters were entered in the sensitivity analysis?

*Response Q(1): Of course, we are happy to clarify the parameters of our sensitivity analysis (lines 317-319) :

“We also conducted a sensitivity analysis in G*power (49), setting power to .80 and the probability of alpha-error to .05.“

Q(2): Presentation order of the task

What was the order in which the tasks were presented at each time point? Did all the children perform the task in the same order or was the presentation order of tasks counterbalanced? This could be clarified in the procedure.

*Response Q(2): As suggested, we revised the paragraph “Procedure” to clarify the order of task administration. We have now clarified that all children performed the tasks in the same order (lines 420-422). We added the following information:

“Task order was fixed across all timepoints. Children completed the magnitude comparison tasks before receiving the non-symbolic and symbolic order processing tasks.“

Q(3): Items on the order tasks

With regard to the non-symbolic order processing and symbolic order processing task, it is mentioned that there were 41 ascending triplets for the non-symbolic task and 35 ascending triplets for the symbolic task among 80 items. Were the remaining trials all non-ordered? In the last paragraph about the limitations of the study, it is mentioned that no descending trials were excluded, so I assume that only ascending and non-ordered trials were included but this could be clarified in the method section. Furthermore, I wondered why the authors did not choose to present half of the items in ascending order since an uneven distribution of items in the different conditions could have led to a response bias.

*Response Q(3): Thank you for asking about the different conditions of the order processing tasks. Reviewer #1 also asked us to mention in the task description (and not only in the limitations) that we did not include descending items. We have revised the task description to reflect a) that the remaining XY trials were not in order and b) that we did not include descending items as we found them to be too difficult for children in Grade 1 (lines 323-234).

Concerning the slightly unequal amount of ascending and not-in-order items, we do not think that this led to a response bias. These were timed tasks and none of the participants managed to solve all 80 items within the given time limit of 90 seconds. As a matter of fact, depending on how far they got, some children may have encountered and equal or an unequal number of ordered and non-ordered items. In an experiment comparing performance across conditions, it would of course have been critical to balance items across conditions.

Q(4): Mean performance on the tasks

It would be informative to not only report the descriptive statistics for the order tasks, but also for the other tasks such as the working memory tasks and counting task. In addition, I wondered whether there were any outliers for the symbolic order processing task, working memory tasks and counting task. Outliers have been reported for the non-symbolic order processing task but not for the other tasks.

*Response Q(4): Descriptive statistics for the predictors are displayed in Table 2, together with zero-order correlations. We hope we have made this clearer by also mentioning this in the text in the following way:

“Zero-order correlations between the predictors as well as descriptive statistics are shown in Table 2.” (line 477)

We only reported outliers for the small sample for the separate investigation of retest-reliability. For our main study, we did not exclude any outliers. Indeed, apart from children with incomplete data, we only excluded children with a clearly biased answering tendency in either of the order processing tasks (e.g., ticking or crossing out more than 10 items in a row) (cf. lines 307-309)

Q(5):The predictive role of verbal and visual-spatial working memory

Could the finding that visuospatial working memory was a strong predictor for order processing be specific for younger children? Literature about the development of arithmetic performance shows that visual-spatial working memory is especially a strong predictor when children are not familiar yet with the arithmetic operations. When arithmetic operations become more familiar, verbal working memory plays a more important role (see Ragubar et al., 2010 for a review). A similar process could play a role in the development of order processing and verbal working memory could become more important for order processing as children acquire a better understanding of ordered sequences. In line with this reasoning, the authors do find a significant correlation between verbal manipulation and symbolic ordering for time point 2 and 3 while they did not observe this correlation for time point 1.

*Response Q(5): Thank you for raising this point! We did not previously consider this, but we believe that discussing this possibility improves our line of argumentation. We thus extended our discussion on the role of visuo-spatial working memory (lines 673-677) and verbal working memory (lines 689-710).

---

## [Decision Letter · Decision Letter 1]

7 Oct 2021

Common and distinct predictors of non-symbolic and symbolic ordinal number processing across the early primary school years

PONE-D-21-14700R1

Dear Dr. Finke,

We’re pleased to inform you that your manuscript has been judged scientifically suitable for publication and will be formally accepted for publication once it meets all outstanding technical requirements.

Kind regards,

Jérôme Prado

Academic Editor

PLOS ONE

Additional Editor Comments (optional):

Reviewers' comments:

Reviewer #1: I would like to thank the authors for considering all of the comments. I thought the manuscript was already of very good quality. I think it is now even more readable and understandable. In addition, it is a real and important contribution to the field.

Reviewer #2: This study sheds important lights on the predictors of symbolic and nonsymbolic order processing in primary school. In their revision, the authors took great care of addressing each of the previous concerns and in their cover the authors explained clearly how they revised their manuscript. The manuscript substantially improved by elaborating on some studies in the introduction and by further reflecting on the findings in the discussion. Furthermore, the authors clarified important points of the methodology of the study. A final comment that I have is that I did not find it straightforward to find the data described in the manuscript through the doi and link which the authors provided. The link directs to the data of a larger project and only a part of this data is analysed in the current study. Therefore, it would be helpful to give a short description to the reader about which files are related to the data described in the manuscript.

7. PLOS authors have the option to publish the peer review history of their article (what does this mean?). If published, this will include your full peer review and any attached files.

Reviewer #1: **Yes: **Sandrine Mejias

Reviewer #2: No

---

## [Editor Report · Acceptance letter]

12 Oct 2021

PONE-D-21-14700R1 

Common and distinct predictors of non-symbolic and symbolic ordinal number processing across the early primary school years 

Dear Dr. Finke:

I'm pleased to inform you that your manuscript has been deemed suitable for publication in PLOS ONE. Congratulations! Your manuscript is now with our production department. 

Kind regards, 

on behalf of

Dr. Jérôme Prado 

Academic Editor

PLOS ONE